# Can Coagulation System Disorders and Cytokine and Inflammatory Marker Levels Predict the Temporary Clinical Deterioration or Improvement of Septic Patients on ICU Admission?

**DOI:** 10.3390/jcm10081548

**Published:** 2021-04-07

**Authors:** Georgia-Athanasia Lavranou, Spyros Mentzelopoulos, Paraskevi Katsaounou, Ilias Siempos, Ioannis Kalomenidis, Aikaterini Geranaki, Christina Routsi, Spyros Zakynthinos

**Affiliations:** 1First Department of Intensive Care Medicine, School of Medicine, National and Kapodistrian University of Athens, ‘Evangelismos’ Hospital, 45-47 Ipsilandou St, GR-10675 Athens, Greece; atlavranou@gmail.com (G.-A.L.); sdmentzelopoulos@gmail.com (S.M.); paraskevikatsaounou@gmail.com (P.K.); isiempos@yahoo.com (I.S.); ikalom@med.uoa.gr (I.K.); chroutsi@hotmail.com (C.R.); 2Hematology Laboratory, ‘Evangelismos’ Hospital, 45-47 Ipsilandou St, GR-10675 Athens, Greece; kategeran@gmail.com

**Keywords:** sepsis, septicshock, coagulation, coagulation inhibitors, procalcitonin, thrombopoietin, cytokines, antithrombin III, protein C

## Abstract

Although coagulation disorders and immune/inflammatory response have been associated with the final outcome of patients with sepsis, their link with thetemporaryclinical deterioration or improvement of patients is unknown. We aimed to investigate this link. We prospectively included consecutive patients admitted to the intensive care unit (ICU) with a suspected diagnosis of infection and evaluated within the first 24 h from admission. Blood levels of many cytokines and inflammatory and coagulation factors were measured and their predictive value was assessed by calculating the Area Under the Receiver Operating Characteristic (AUROC) curves. Patients (*n* = 102) were allocated in five groups, i.e., sepsis (*n* = 14), severe sepsis (*n* = 17), septic shock (*n* = 28), Systemic Inflammatory Response Syndrome (SIRS) without infection (*n* = 17), and trauma/surgery without SIRS or infection (*n* = 26). In septic shock, coagulation factors FVII and FIX and Protein C had AUROCs 0.67–0.78. In severe sepsis, Antithrombin III, Protein C, C-reactive protein, Procalcitonin and Thrombopoietin had AUROCs 0.73–0.75. In sepsis, Tumor Necrosis Factor a, and Interleukins 1β and 10 had AUROCs 0.66–0.72. In patients admitted to the ICU with a suspected diagnosis of infection, coagulation factors and inhibitors, as well as cytokine and inflammatory marker levels, have substantial predictive value in distinct groups of septic patients.

## 1. Introduction

Sepsis is one of the leading causes of death for patients in the Intensive Care Unit (ICU) and a major contributor to the growing financial burden of medical care worldwide [1,2]. As a series of inflammatory and homeostatic changes that occur as a reaction to systemic infection, sepsis is defined as the suspected or proven infection and coexisting Systemic Inflammatory Response Syndrome (SIRS: fever, tachycardia, tachypnea, leukocytosis, etc.) [3,4,5,6]. Severe sepsis is defined as sepsis in combination with organic dysfunction (hypotension, hypoxemia, metabolic acidosis, thrombocytopenia, etc.) [3,4,5,6]. Septic shock is defined as severe sepsis in combination with hypotension despite the adequate recovery of body fluids [3,4,5,6].

In 2016, a new definition of sepsis was created (Sepsis-3), according to which sepsis is defined as an infection that causes organic dysfunction, with the abolition of the term SIRS when referring to sepsis and the term severe sepsis [7,8,9]. However, there are conflicting views in the literature as to the necessity of the new definition and objections, in particular to the abolition of the term SIRS [10]. Nevertheless, we will use the old definitions in this study.

Septic shock and multiorgan dysfunction are the most common causes of death in patients with sepsis [3,4,5,6]. Several systemic factors that interact to promote organic deficiency have been evaluated in many studies, suggesting that coagulation disorders occur even before the onset of clinical symptoms of severe sepsis or septic shock, are associated with the severity of the disease, and are likely to predict mortality [2,5,6,11]. Additionally, pro-inflammatory cytokines [12] and other molecules, like the hormone thrombopoietin (TPO) [13], have been systematically screened, considered biomarkers of severity, and possibly able to predict the final outcomein the sense of survival or death of the septic syndrome. However, to the best of our knowledge, it is unknown whether the various disorders of the coagulation system or the immune and inflammatory response can predict the temporary clinical deterioration or improvement of the patients admitted to the ICU due to suspected infection.

The present study aimed to investigate whether the various disorders of the blood coagulation system and the immune and inflammatory response (i.e., levels of pro- and anti-inflammatory cytokines and other biomarkers of sepsis) can predict the temporary worsening or improvement of the clinical status of patients admitted to the ICU due to suspected infection. Therefore, we included patients admitted to the ICU and evaluated them on admission and then daily until their discharge from the ICU or death.

## 2. Materials and Methods

### 2.1. Patients and Protocol

This research study was conducted in the ICU of the General Hospital “Evangelismos”, Intensive Care Clinic of the Medical School of the National and Kapodistrian University of Athens. Over a period of one and a half years (1 January 2016–30 June 2017), were prospectively included all consecutive patients who were admitted to the ICU with a suspected diagnosis of infection, were evaluated within the first 24 h from admission and finally confirmed that: (a) had sepsis, severe sepsis or septic shock, or (b) had a disease associated with SIRS without infection. Additionalwere consecutively included patients who had undergone trauma or surgery without suspected infection and did not meet the SIRS criteria, but could potentially develop SIRS and sepsis. The latter group was studied as a control group. Were excluded patients: (a) under 16 years of age; (b) with chronic hepatic failure, chronic renal failure, known hematological disease with coagulation or platelet count disorders;(c) with malignancy under chemotherapy or malignancies associated with coagulation or platelet count disorders; (d) under anticoagulant or antiplatelet therapy; and (e) who had been transfused with units of fresh frozen plasma or concentrated coagulation agents or platelet units within the previous 48 h of admission to the ICU.Thestudywasapprovedbythehospitalinstitutionalreviewboards (ethical committee of “Evangelismos” Hospital; 76/14 February 2013). Informed consent was waived.

Blood levels of the following cytokines and inflammatory and coagulation factors were measured in the first 24 h after admission to the ICU: Procalcitonin (PCT), C-reactive protein (CRP), TPO, Interleukin 6 (IL-6), Tumor Necrosis Factor a (TNF-a), Interleukin 1β (IL-1b), Interleukin 10 (IL-10), Antithrombin III (ATIII), Protein C (PrC), D-dimmers (Dds), Fibrinogen (Fibrin), Plasminogen, and coagulation factors FV, FVII, FVIII, FIX, FvWillebrand (vWF), and FX.

In the first 24 h after admission to the ICU, were also measured the arterial blood gases and usual laboratory parameters [white blood cells (WBC), red blood cells (RBC), hematocrit (Ht), platelets (PTL), creatinine (Creatin), bilirubin (Bil), prothrombin time (PT), activated partial thromboplastin time (a-PTT), lactic acid (Lactate), glucose (Glu), and albumin (Alb)]. Cultures of biological fluids were obtained in suitable transport media to isolate potential pathogenic microorganisms. Imaging was performed on admission (at least chest X-ray) and where specific diagnostic tests (ultrasound, computed tomography) were required to document a possible source of infection.All these clinical measures have scientific rationale based on the current mechanistic understanding of sepsis and were taken by the attending physicians according to the guidelines used in our ICU [6].

All included patients were allocated in five groups, i.e., sepsis, severe sepsis, septic shock, SIRS without infection, and trauma or surgery without SIRS or infection. The classification of patients in one of these groups took place in the days following the admission of patients when the results of laboratory, bacteriological, and imaging tests were completed. This classification, as well as the clinical assessment of patients from ICU admission until their exit from the ICU or death, was performed by two experienced ICU physicians who did not participate in the study. These physicians assessed the stage of sepsis based on ACCP/SCCM criteria [3], the severity of the general condition of the patient with APACHE II score (range 0–71, with higher scores indicating more severe disease and a higher risk of death) [14], the presence of Multiple Organ Dysfunction Syndrome (MODS) (with yes indicating dysfunction/failure of at least 2 organs) [15], the severity of organ dysfunction with Sequential Organ Failure Assessment (SOFA) score (range 0–24, with higher scores indicating more severe illness) [16] and the severity of acute lung injury with the Lung Injury Score (LIS) [with 0 indicating the absence of acute lung injury, 0.1–2.5 indicating the presence of mild-to-moderate acute lung injury, and >2.5 indicating the presence of Acute Respiratory Distress Syndrome (ARDS)] [17]. Additionally, these two ICU doctors assessed the location and the cause of sepsis. Subsequently, a daily clinical and laboratory evaluation of the patients was performed until their discharge from the ICU or death. The aim was to determine the improvement or deterioration of the patients’ clinical condition in order to assess the predictive value of the coagulation components and the immune and inflammatory response measured in the first 24 h after the patients entered the ICU.

Clinical improvement was defined as discharge from the ICU, a decrease of at least 1 point from baseline on a six-point ordinal scale, or both. The six-point scale consisted of the following categories: (1) not in the ICU; (2) in the ICU, without SIRS or infection; (3) in the ICU, with SIRS or sepsis; (4) in the ICU, with severe sepsis; (5) in the ICU, with septic shock; and (6) death. Clinical deterioration was defined as death, an increase of at least 1 point from baseline on the six-point ordinal scale, or both. The first change in the clinical status (i.e., improvement or deterioration) of the patients was taken into account for the evaluation of the predictive value of the variables measured within the first 24 h from admission.

### 2.2. Sample Collection and Laboratory Analysis

Blood samples were obtained by collecting whole blood from a peripheral vein, on the one hand for the measurement of cytokines and inflammatory markers in sterile Wasserman tubes and collection of serum after whole blood coagulation for 45–60 min, and on the other hand for the measurement of factors of coagulation in tubes containing 3.8% sodium citrate as an anticoagulant for plasma collection. Both serum and plasma were taken up after centrifugation for 10 min at 3000 rpm. Serum and plasma were used immediately or kept frozen at −80 °C for subsequent analysis in 0.5 mL plastic tubes. All analyzes were performed without knowing either the patient’s name or the day of sample collection.

#### 2.2.1. Determination of Inflammatory Markers

The acute phase protein identified in our study was CRP measured by a nephelometric technique (BN 100; Medgenix Diagnostics, Fleurius, Wevelgem, Belgium; normal values: <1.0 mg/dL). PCT serum levels were measured using an immunoassay with a sandwich technique and a chemiluminescent detection system, according to the manufacturer’s protocol (LumiTest; Brahms Diagnostica, Berlin, Germany; normal values: <0.1 ng/mL). TPO serum levels were measured using an enzyme-linked immunosorbent assay (Quantikine; R&D Systems, Abingdon, UK; normal values: 33–310 pg/mL).

#### 2.2.2. Determination of Cytokines

The determination of cytokines TNF-a, IL-1b, IL-10, and IL-6 in the serum was done by the ELISA (enzyme-linked immunosorbent assay) method, using commercial diagnostic kits (Quantikine; R&D Systems, Abingdon, UK; normal values: 0.2–6.3 pg/mL, <3.0 pg/mL, <5.0 pg/mL and <2.0 pg/mL, respectively).

#### 2.2.3. Determination of Coagulation Factors

The activity of ATIII, PrC and Plasminogen was measured in plasma by a chromogenic assay (Dade–Behring; normal values: 80–120%, 70–130% and 70–130%, respectively). The activity of coagulation factors FV, FVII, FVIII, FIX, vWF, and FX was determined by the method of micro-ELISA (Dade–Behring; normal values: 50–200%, 70–120%, 70–140%, 70–120%, 50–120% and 70–120%, respectively). The determination of Fibrinogen was done with a modification of the classic Clauss coagulation method (Dade–Behring; normal values: 150–400 mg/dL). Dds were measured by an immuno-turbidimetric assay (DiagnosticaStago, France; normal values: <0.30 μg/mL).

### 2.3. Statistical Analysis

The values of the various quantitative variables were expressed as median and range values due to the absence of a normal value distribution (evaluation was done by the Kolmogorov—Smirnov good fit test). Therefore, non-parametric tests were used for group comparisons. Differences between multiple groups were assessed by the Kruskall Wallis analysis of Variance by ranks, while comparisons between two groups were made by the Mann–Whitney U test. Differences in qualitative variables were assessed using the Yates corrected Chi-Square test. Differences with *p*-value <0.05 were considered statistically significant. The evaluation of the predictive value of the various variables was done by creating ROC (Receiver Operating Characteristic) curves. Areas below the ROC curve (Area under the ROC, AUROC) were considered poor when they were 0.6–0.7, sufficient 0.7–0.8, good 0.8–0.9, and excellent when they were ≥0.9 [18]. The statistical analyzes were performed with the software SPSS Statistics version 17.0.

## 3. Results

### 3.1. Clinical and Laboratory Characteristics of Patients on ICU Admission

The study included prospectively 102 patients: 26 patients with trauma or surgery who did not meet at least two of the SIRS criteria, 17 patients with SIRS (6 with multiple trauma, 4 with cerebral injury, 3 with ARDS [19], 2 with cardiovascular shock and 2 with hemorrhagic shock), 14 patients with sepsis (11 with pneumonia and 3 with other foci of infection), 17 patients with severe sepsis (13 with pneumonia, 2 with intra-abdominal infection and 2 with other foci of infection) and 28 patients with septic shock (17 with pneumonia, 8 with intra-abdominal infection and 3 with other foci of infection). Sixty-two (61%) patients were male. The treatment of patients was decided by the attending physicians according to the guidelines used in our ICU [6], and included administration of antibiotics, appropriate fluid management, etc.Some of the comorbidities (e.g., chronic lung disease) are incorporated in the chronic health point system of the APACHE II score. In more detail, 28/102 patients (27%) had hypertension, 14/102 (14%) had diabetes mellitus, 7/102 (7%) had coronary artery disease, 8/102 (8%) had congestive heart failure, 10/102 (10%) had obesity, defined as Body Mass Index > 30 kg/m^2^, and 3/102 patients (3%) had chronic lung disease, whereas 39/102 patients (38%) had at least two comorbidities.The demographic, clinical and usual laboratory data of patients upon admission to the ICU are presented in Table 1. As expected, patients with septic shock were more severe than patients in the other groups. Indeed, patients with septic shock had higher APACHE and SOFA scores, LIS, MODS, lactate, creatinine, and bilirubin, as well as lower PaO_2_/FiO_2_ ratio and platelet count. Infections were microbiologically proven in 50 of the 59 patients with infection (85%), of which 43 (86%) were due to a Gram-negative germ, 3 (6%) to a Gram-positive germ and 4 (8%) were mixed infections.

### 3.2. Measurements of the Coagulation Components, Cytokines, and Inflammatory Markers of Patients on ICU Admission

Measurements of the coagulation system components, cytokines, and inflammatory markers are presented in Table 2. In patients with severe sepsis or septic shock, PT and Dds were higher than in patients with trauma/surgery and SIRS, while Dds were also higher than in patients with uncomplicated sepsis.

Significant differences between the study groups were observed in the activity of coagulation factors FVII, vWF, FX, and FV, as well as in the levels of cytokines TNF-a and IL-1b, inflammatory markers CRP and PCT, and hormone TPO.

### 3.3. Predictive Value, in the Sense of Patient Temporary Clinical Deterioration or Improvement, of the Various Factors and Variables Measured on ICU Admission

Comparisons of the values of the various factors and variables between patients who deteriorated or improved in the five patient groups of the study are presented in Table 3, Table 4, Table 5, Table 6 and Table 7. Table 8 presents the predictive value of the various factors and variables, in the sense of temporary clinicaldeterioration or improvement, through calculation of the AUROCs (95% confidence interval) of these factors and variables that showed a statistically significant difference in the patient group comparisons of Table 3, Table 4, Table 5, Table 6 and Table 7. Figure 1 illustrates three representative ROC curves, one for each septic group (sepsis, severe sepsis, and septic shock).

The treatment was not different between patients who deteriorated or improved in all five patient groups.Moreover, the percentages of the various comorbidities were not significantly different between patients who deteriorated or improved in all five patient groups.

#### 3.3.1. Clinical and Laboratory Variables

The PaO_2_/FiO_2_ ratio was significantly lower in patients who deteriorated than in those who improved in all patient groups except sepsis. The predictive value of the PaO_2_/FiO_2_ ratio was poor in the trauma/surgery and SIRS patient groups, while it was excellent for the severe sepsis group and sufficient for the septic shock group. APACHE, SOFA, and LIS were significantly higher in patients who worsened than in those who improved in all three groups of septic patients. The predictive value of all three scores was sufficient or good. The rate of MODS was significantly higher in patients who worsened than in those who improved only in the group of patients with septic shock where its predictive value was excellent. Age was significantly higher in patients who deteriorated than in those who improved only in the group of patients with severe sepsis albeit with poor predictive value.

Among the usual laboratory parameters, only blood lactic acid was significantly higher in patients who worsened than those who improved solely in the group of patients with septic shock where its predictive value was good.

#### 3.3.2. Coagulation System, Cytokines, and Markers of Inflammation

PT and Dds were significantly higher in patients who worsened than in those who improved only in the trauma/surgery group where their predictive value was sufficient. Among the coagulation factors, activities of FVII and FIX were significantly lower in patients who worsened than in those who improved only in the group of patients with septic shock with sufficient and poor predictive value, respectively. Among the other coagulation factors, only the activity of FVIII was significantly lower in patients who worsened than in those who improved in the trauma/surgery group albeit with poor predictive value. Among the coagulation inhibitors, ATIII activity was significantly lower in patients who deteriorated compared to those who improved in the group of patients with severe sepsis with sufficient predictive value. PrC activity was also significantly lower in patients who worsened than in those who improved in the groups of patients with severe sepsis and septic shock with sufficient predictive value (please see also Appendix A).

Among the cytokines, TNF-α levels were significantly higher in patients who worsened than in those who improved in the trauma/surgery and sepsis patient groups with sufficient and poor predictive value, respectively. IL-1b and IL-10 levels were significantly higher in patients who worsened than in those who improved only in the group of patients with sepsis with sufficient predictive value.

CRP, PCT, and TPO levels were significantly higher in patients who deteriorated than in those who improved only in the group of patients with severe sepsis with sufficient predictive value.

## 4. Discussion

Despite many pre-clinical studies and clinical trials, there is no known treatment forsepsis. Based on the assessment of the various factors at admission, the present study aimed to unravel the most critical factors for the outcome of sepsis, and thus point to the type of intervention needed, or to the direction of scientific/clinical research to eventually improve the outcome.Our main findings were: (1) Among the coagulation system factors, only a few had a predictive value. In particular, FVII and FIX activities were lower in patients who deteriorated compared to those who improved only in the group of patients with septic shock with sufficient (AUROC 0.72) and poor (AUROC 0.67) predictive value, respectively. Among the other coagulation factors, only FVIIII activity was lower in patients who worsened than in those who improved in the trauma/surgery group albeit with poor predictive value (AUROC 0.63); (2) Among the coagulation inhibitors, both ATIII and PrC had significant predictive value. In particular, ATIII activity was lower in patients who deteriorated than in those who improved in the group of patients with severe sepsis with sufficient predictive value (AUROC 0.74). PrC activity also was lower in patients who deteriorated compared to those who improved in the groups of patients with severe sepsis and septic shock with sufficient predictive value (AUROCs 0.75 and 0.78, respectively); (3) Cytokine levels had significant predictive value mainly in the group of patients with sepsis. Indeed, in this group, TNF-α, IL-1b and IL-10 levels were higher in patients who worsened than in those who improved with poor to sufficient predictive value (AUROCs 0.66, 0.71 and 0.72, respectively); (4) CRP, PCT, and TPO levels had significant predictive value only in the group of patients with severe sepsis. Indeed, in this group, the levels of all these proteins were higher in patients who worsened than in those who improved with sufficient predictive value (AUROCs 0.73–0.75); (5) Among the usual laboratory variables, only blood lactic acid had a significant predictive value restricted in the group of patients with septic shock. Indeed, lactate levels were higher in patients who deteriorated compared to those who improved in septic shock with good predictive value (AUROC 0.87); and (6) The various clinico-laboratory scores and measurements had a significant predictive value in almost all groups of septic patients. Specifically, APACHE, SOFA, and LIS were higher in patients who deteriorated compared to those who improved in all three groups of septic patients with sufficient to good prognostic value (AUROCs 0.72–0.84), PaO_2_/FiO_2_ ratio was lower in patients who worsened than in those who improved in the groups of patients with severe sepsis and septic shock with excellent (AUROC 0.90) and sufficient (AUROC 0.79) predictive value, respectively, while the rate of MODS was higher in patients who worsened compared to those who improved only in the group of patients with septic shock where its predictive value was excellent (AUROC 0.91).

### 4.1. Usual Laboratory Parameters and Severity Scores

#### 4.1.1. SOFA Score

The SOFA rating index is a simple but effective method to describe organ dysfunction in severely ill patients. It was originally designed to evaluate/describe and not to predict the survival expectancy of seriously ill patients [20]. It includes the evaluation of six organ-systems with a score of 0–4 for each organ-system. Systematic and repetitive scoring helps to better monitor and understand the clinical picture of patients [20]. SOFA score does not work only for septic patients, and the European-North American Study of Severity System database showed a satisfactory correlation of the SOFA score with survival [16,21]. Indeed, in a study by Vincent et al., data were collected from 1449 critically ill patients in 40 ICUs and the SOFA score was found to be satisfactorily related to survival [16]. Respiratory failure was more common than other organ dysfunctions and was a very sensitive parameter. Thus, patients with respiratory failure had a higher SOFA score in a shorter period of time than patients with hepatic impairment. This has been attributed to the fact that the increase in bilirubin takes time and may therefore lead to liver failure being recognized later [16]. Another study evaluated the mean and highest value of the SOFA score as prognostic indicators of survival; regardless of the initial value, an increase in the SOFA score in the first 48 h after admission to the ICU is a predictor of mortality of at least 50% [20].

In our study, when patients entered the ICU, as expected, those with septic shock were more severely ill than patients in the other groups and had higher SOFA scores (Table 1). SOFA score also had a very significant predictive value in all groups of septic patients. Indeed, the SOFA score was higher in patients who worsened compared to those who improved in all three groups of septic patients (Table 5, Table 6 and Table 7) with sufficient to good predictive value (AUROCs 0.72–0.82) (Table 8).

#### 4.1.2. APACHE II Score

APACHE II score is widely used in ICUs as a system for assessing the severity of patients, with high values associated with increased mortality [14]. An earlier study comparing the initial APACHE II values of patients entering the ICU with the worst values of the first 24 h, showed that the two scores in critically ill non-injured patients did not differ in their predictive capacity [22]. In patients with ventilator-associated pneumonia, APACHE II appeared to be the most reliable tool for predicting mortality compared to other suggested scores [23]. In a relatively recent study [24], the ability of APACHE II to predict in-hospital mortality in critically ill patients declines over the years, leading authors to suggest a possible renewal of some of its parameters. In contrast, other researchers argue that it remains useful to differentiate patients by their severity using APACHE II [25].

In our study, as expected, patients with septic shock were more severely ill than patients in the other groups, and had a higher APACHE II score (Table 1). Additionally, the APACHE II score had a significant predictive value in all groups of septic patients. Indeed, APACHE II was higher in patients that worsened than in those who improved in all three groups of septic patients (Table 5, Table 6 and Table 7) with sufficient to good predictive value (AUROCs 0.76–0.84) (Table 8).

#### 4.1.3. PaO_2_/FiO_2_ Ratio and Lung Injury Score

On ICU admission, both PaO_2_/FiO_2_ ratio and LIS could predict the need for mechanical respiratory support, but PaO_2_/FiO_2_ ratio was a better prognostic indicator for the length of stay in the ICU compared to the LIS [26]. LIS has also been used as a predictor of mortality with higher values associated with increased mortality [17,27].

In our study, when entering the ICU, patients with septic shock had a more severe degree of respiratory failure than patients in the other groups (higher LIS and lower PaO_2_/FiO_2_ ratio) except those in the severe sepsis group (Table 1). LIS also had a significant predictive value in all groups of septic patients. Indeed, LIS was higher in patients who deteriorated compared to those who improved in all three groups of septic patients (Table 5, Table 6 and Table 7) with sufficient to good predictive value (AUROCs 0.78–0.82). Furthermore, PaO_2_/FiO_2_ ratio had a significant predictive value for the groups of patients with severe sepsis and septic shock. Indeed, PaO_2_/FiO_2_ ratio was lower in patients who worsened than in those who improved in the groups of patients with severe sepsis and septic shock (Table 6 and Table 7) with excellent (AUROC 0.90) and sufficient (AUROC 0.79) predictive value, respectively (Table 8).

#### 4.1.4. Lactic Acid

The relationship between elevated blood lactic acid levels and tissue hypoxia has been noted since 1927 in patients with shock [27]. Several experimental and clinical studies have shown that lactate levels increase in tissue hypoxia [28]. Moreover, elevated lactate levels are sufficient to diagnose shock regardless of hypotension [29], and lactate levels are indicators of mortality rate in patients with trauma and sepsis [30]. Sepsis with lactate levels ≥ 4 mmol/L is associated with high mortality and is an indication for initiation of treatment protocols [31]. Meregalli et al. [32] showed that in postoperative patients with similar hemodynamic parameters blood lactate levels in the first 12 h after ICU admission are those that will predict survival. Changes in lactate levels over time can be a predictor of survival and show a response to treatment [32]. Vincent et al. described changes in lactate levels over time after resuscitation in patients with circulatory shock and showed that those patients who died did not present a decrease in baseline lactate levels after resuscitation [33]. Other authors, studying only patients with multiple injuries, showed that the improvement of hemodynamic parameters, i.e., cardiac output, oxygen consumption, and oxygen supply, are not predictive indicators of survival, whereas the optimization of blood lactate levels is a prognostic indicator of survival [34].

In the present study, at the time of ICU admission, patients with septic shock had higher blood lactate levels than patients in the other groups (Table 1). Lactate levels had a significant predictive value for the group of patients with septic shock. Indeed, lactate levels were higher in patients who deteriorated compared to those who improved in septic shock (Table 7) with good predictive value (AUROC 0.87) (Table 8).

### 4.2. Coagulation System

Coagulation disorders are strongly linked to the process of sepsis. For example, fibrinolysis which involves a complex system of activation and inhibition mechanisms is affected during sepsis so that the result is reduced fibrinolysis, deposition of microthrombi in the vascular bed, and multiorgan failure [35].

#### 4.2.1. Platelets

Platelets play an important role in the normal formation of thrombus-hemostasis. After activation, they change shape to increase their ability to adhere by activating glycoprotein receptors on their surface [36]. Activated platelets secrete various proteins including oxidizing agents, platelet-activating agents, complement proteins, cytokines, and other enzymes that modulate their action but also affect the action of the cells to which they attach (endothelium and neutrophils) [36]. However, platelet activation also has potentially detrimental effects. Platelet aggregation in the area of inflammation may be responsible for microcirculation disorders, thus contributing to organ dysfunction and insufficiency in patients with sepsis [37,38]. Of course, their primary role is to activate the defense mechanisms that will contribute to healing in the area of the lesion and vascular remodeling [38,39].

An acute decrease in platelet count occurs in the early stages of many diseases. It is due to various causes such as reduced production, increased consumption, or pathological fragmentation [40]. Their reduced production may be due to suppression of the bone marrow by infectious agents, toxic drugs or mediators of inflammation. Their increased destruction is often a side effect of drugs, such as heparin which through immunostimulation can reduce the half-life of platelets [41]. Injured or postoperative patients lose circulating platelets, thus showing thrombocytopenia in severe cases. Patients, especially after cardiac surgery, have platelet dysfunction after the extracorporeal circulation which they undergo [42]. Thrombocytopenia is common in severely ill patients and has been associated with a worse prognosis in several studies [38,39]. Patients with ARDS may have reduced platelet counts due to their entrapment in the lungs [43], while patients with diffuse intravascular coagulation have a high consumption of platelets and coagulation factors from the microvascular network of many organs [43]. Thrombocytopenia in patients admitted to the ICU is an indicator of poor prognosis [44] and is associated with a longer stay in the ICU [40]. Akca et al. [42] reported that thrombocytopenia in septic patients had a relative risk for death of 1.66 while Brun–Buisson et al. [44] found a relative risk of 1.5 in patients with platelet counts <50.000/mL.

Although thrombocytopenia in an ICU has been associated with worse survival expectancy, the exact correlation between the change in platelet count over time and the mortality rate has not been established [42,44]. Akca et al. [42] showed changes in platelet counts in severely ill patients with a biphasic pattern that differed in those who survived from those who died. Late thrombocytopenia was associated with increased mortality compared to early; although thrombocytopenia was more common on the 4th day of hospitalization than on the 14th, the mortality rate was higher in late thrombocytopenic patients [42]. Moreover, in thrombocytopenic patients, an increase in platelet count occurred in surviving patients but was not observed in those who died [42]. In this study, individual platelet counts were of little value in predicting life expectancy, but changes in their number over time correlated with patient life expectancy [42]. Similar biphasic changes in platelet counts have been reported in postoperative patients [45] and myocardial infarction [46], as well as healthy donors after plasmapheresis [47]. Smith-Erichsen showed this biphasic distribution in a small study of 18 surgical patients with severe sepsis [48]. Patients who died had persistent thrombocytopenia while survival was associated with the degree of thrombocytopenia over two weeks [48]. In another study, a large number of ICU patients were evaluated and no correlation was found between platelet count on admission and survival [49]. However, patients who eventually died appeared to have a smaller increase in platelet count between days 2 and 10 than those who survived [49].

In the present study, at the time of ICU admission, patients with septic shock had lower platelet counts than all other groups, and patients with severe sepsis had lower platelet counts than those with SIRS and sepsis (Table 1). No difference in platelet counts was observed between patients who deteriorated compared to those who improved in all groups of the study (Table 3, Table 4, Table 5, Table 6 and Table 7); therefore, the predictive value of platelet counts was negligible.

#### 4.2.2. Prothrombin Time (PT)

PT is often prolonged in septic patients [48]. In the study by Dhainaut et al., changes in PT overtime alone were almost equally capable of predicting mortality at 28 days compared with a combined assessment of Dds and ATIII but had a lower value for the prognosis of multiorgan failure [49].

In our study, on ICU admission, PT was higher in patients with severe sepsis or septic shock than in patients with trauma/surgery and SIRS (Table 2). The predictive value of PT in terms of deterioration or improvement of patients was sufficient only in the group of patients with trauma/surgery (AUROC 0.75), while it was nil in all other groups (Table 8).

#### 4.2.3. Antithrombin III

The activity of the coagulation inhibitor ATIII is frequently low in severely ill patients. This decrease in the activity of ATIII, as well as that of the other coagulation inhibitor PrC, is caused by the combined action of various processes, such as: (a) overconsumption due to increased thrombin production, (b) degradation by plasma elastases, which are released by activated neutrophils, and (c) insufficient synthesis [50,51,52,53,54,55]. Indeed, thrombin is produced and competes with ATIII, resulting in low levels of ATIII in the blood of most patients with severe inflammation [50,51,52,53,54,55]. Along with overconsumption, ATIII is also destroyed by leukocyte proteases [54,55,56]. In general, a hepatic impairment that patients with sepsis may experience affects the coagulation mechanism by reducing the synthesis of coagulation proteins (including coagulation inhibitors) and by reducing the clearance of activated enzymes and complexes of enzymes-inhibitors [50,51,54].

In the present study, on ICU admission, ATIII had lower activity in patients with severe sepsis or septic shock than in the other three groups (Table 2). ATIII activity was lower in patients who worsened than in those who improved in the group of patients with severe sepsis (Table 6) with sufficient predictive value (AUROC 0.74) (Table 8, Figure 1).

#### 4.2.4. Protein C

Besides its action as a coagulation inhibitor, PrC contributes to fibrinolysis, as follows: The conversion of plasminogen to plasmin is activated by tissue-type plasminogen activator (t-PA) and urokinase-like plasminogen activator (uPA). Endothelial cells are the main source of t-PA but t-PA can also be isolated in other tissues [57]. The fibrinolytic process has two levels: initially, these activators can be inhibited by the plasminogen activator inhibitor type 1 (PAI-1) produced by the endothelium and form complexes with them so that they cannot activate the plasminogen [57,58]. The action of PAI-1 is also inhibited by activated PrC which binds and inactivates PAI-1, thus increasing fibrinolysis. In most patients with sepsis or septic shock, PrC activity decreases and is associated with an increased risk of death [11,59,60]. Bernard et al. showed reduced PrC activity (by approximately 50%) in patients with severe sepsis [61].

In the present study, at the time of ICU admission, PrC had lower activity in patients with septic shock than in the other four groups (Table 2). PrC activity was lower in patients who deteriorated compared to those who improved in the groups of patients with severe sepsis and septic shock (Table 6 and Table 7) with sufficient predictive value (AUROCs 0.75 and 0.78, respectively) (Table 8, Figure 1).

#### 4.2.5. Coagulation Factors FVII and FIX

It has been found that the activity of all coagulation factors is gradually reduced in severe sepsis and mainly in septic shock, mainly due to depletion of homeostatic mechanisms [11,60].

In our study, on ICU admission, significant differences in the activity of coagulation factors FVII, vWF, FX, and FV were observed between the groups (Table 2). The activity of FVII and FIX was lower in patients who worsened compared to those who improved only in the group of patients with septic shock (Table 7) with sufficient (AUROC 0.72) and poor (AUROC 0.67) predictive value, respectively (Table 8).

### 4.3. Cytokines

#### 4.3.1. Interleukin 1b, 6 and 10

Many studies have been performed to evaluate interleukins as prognostic markers. In the study of Bozza et al. [62], IL-1b and IL-6 appeared to be the best prognostic markers compared to other cytokines and their prognostic value was much better than the initial evaluation of patients based on the APACHE II score. Several other studies have shown that the majority of patients with sepsis have elevated IL-6 levels and these levels have been associated with severity and survival expectancy [63,64,65]. Constantly elevated IL-6 levels have been associated with multiple organ failure [64] and death [65].

Il-1b is not normally identified in the serum but is detected in the serum of patients with sepsis. McAllister et al. [66] detected IL-1b in the serum of patients who developed sepsis after being transfused with concentrated red blood cells infected with Gram-negative bacteria. These patients had detectable IL-1b that peaked 4 h later and returned to normal in two surviving patients while remained elevated for 22 h in the patient who died. IL-1b is not detected in all septic patients but is an indicator of sepsis severity [67]. Endo et al. [68] found elevated serum IL-1b levels in only 2 of 40 patients with sepsis, but in 15 of 22 patients with septic shock. The findings of our study are consistent with those of Endo et al. [68] because IL-1b levels in patients with septic shock were significantly higher than those in patients with sepsis when patients were admitted to the ICU (Table 2). Goldie et al. detected plasma IL-1b in 29% of 146 patients with sepsis but found no association with mortality [69]. Generally, studies to date show that IL-1b is elevated in the serum of some patients with sepsis and that initial concentrations may be associated with disease severity, but not with mortality [67,68,69].

IL-10 was originally described as an inhibitor of cytokine production by activated macrophages. Gerard et al. [67] showed that administering IL-10 to mice before endotoxin infusion protects against endotoxin-induced mortality and reduces TNF-α production, while other authors [70] showed that administering anti-IL-10 mice antibodies increase TNF-α production and mortality. Several studies have shown that IL-10 is detected in the serum of patients with sepsis. Van Deuren et al. [71] found higher concentrations in patients with septic shock than in septic patients without shock and other authors [68] reported higher concentrations in patients with septic shock than in patients with uncomplicated sepsis; however, these results were not confirmed by the findings of our study (Table 2).

In the present study, IL-1b and IL-10 levels had significant predictive value only in the group of patients with sepsis. Indeed, IL-1b and IL-10 levels were higher in patients who deteriorated compared to those who improved only in the group of patients with sepsis (Table 5) with sufficient predictive value (AUROCs 0.71 and 0.72, respectively) (Table 8, Figure 1).

#### 4.3.2. TNF-a

TNF-a is a precursor of inflammation in a large number of inflammatory diseases, infectious and non-infectious [69]. TNF-a can be detected in the serum of many patients with sepsis and its concentrations are correlated with both severity and prognosis. Endo et al. [68] showed that the serum concentrations of TNF-a, Il-1b, and Il-6 in patients with septic shock were higher than those in patients with uncomplicated sepsis, or with shock from other causes. The findings of our study are consistent with those of Endo et al. [68] because TNF-a levels of patients with septic shock were significantly higher than those of patients with sepsis upon admission to the ICU (Table 2). Casey et al. [72] showed that TNF-a, Il-1b, and Il-6 levels in patients with sepsis may have been higher compared to those in non-sepsis ICU patients, but TNF-a levels alone had no predictive value in terms of mortality. In other studies, elevated TNF-a concentrations were associated with a worse prognosis in patients with sepsis [64,73]. Martin et al. [73] repeatedly measured TNF-a and Il-6 in patients with septic shock and showed that non-surviving patients had consistently higher TNF-a levels compared with those in survivors. Other authors [64] reported that TNF-a concentrations were higher in patients with septic shock compared to non-septic shock and that constantly increased serum TNF-a concentrations predict a worse outcome in patients with shock. Overall, persistentlyincreased concentrations of TNF-a appear to have a better predictive value for survival than individual measurements.

In our study, TNF-a levels were higher in patients who deteriorated compared to those who improved only in the groups of patients with trauma/surgery and sepsis (Table 3 and Table 5) with sufficient (AUROC 0.76) and poor (AUROC 0.66) predictive value, respectively (Table 8).

### 4.4. Inflammatory Markers

PCT in the blood of healthy individuals has values <0.1 ng/mL. In bacterial as well as fungal infections, PCT levels are found to be elevated to some degree, depending on the severity of the infection. Thus, in septic patients, PCT levels may increase 5.000–10.000-fold, while calcitonin levels remain within normal limits [74,75]. In SIRS due to serious and dangerous infections, such as severe sepsis or septic shock, serum PCT levels are particularly high, in contrast to SIRS due to non-infectious causes where PCT levels are usually low. A significant number of studies confirm that PCT is an indicator of serious infection and sepsis. Patients with PCT levels ≤0.5 ng/mL are unlikely to have severe sepsis or septic shock, while levels >2 ng/mL are found in patients at high risk for sepsis or septic shock [76,77]. These findings [74,75,76,77] agree with those of our study. Indeed, at the time of admission to the ICU, PCT levels of patients with septic shock or severe sepsis were significantly higher than those of patients with trauma/surgery, SIRS or uncomplicated sepsis (Table 2).

TPO is a glycoprotein hormone that regulates the number of circulating platelets by stimulating the growth and maturation of megakaryocytes [78]. It is also involved in the later stages of erythropoiesis and induces the proliferation of CD34^+^ progenitor cells [79]. TPO is produced mainly in the liver and secondarily in the kidney by non-hematopoietic cells. Elevated TPO levels have been detected in septic patients with or without diffuse intravascular coagulation; TPO levels may not be inversely related to platelet counts, which reinforces the view that inflammatory cytokines are involved in TPO regulatory mechanisms [13]. Moreover, the correlation of TPO levels with the severity of sepsis has already been reported [13]. These findings [13] are consistent with those of the present study since TPO levels in patients with septic shock or severe sepsis were significantly higher than that of patients with trauma/surgery, SIRS or uncomplicated sepsis (Table 2).

Nevertheless, in the present study, the levels of CRP, PCT and TPO upon admission to the ICU were higher in patients who deteriorated compared to those who improved only in the group of patients with severe sepsis (Table 6) with sufficient predictive value (AUROCs 0.73–0.75) (Table 8).

### 4.5. Strengths and Limitations

This study has several strengths. First, we included all consecutive patients who were admitted to the ICU of a big referral Hospital with a suspected diagnosis of infection, thereby minimizing selection bias. Second, all data were prospectively collected and patients were evaluated on admission and then daily until their discharge from the ICU or death by two independent and experienced ICU physicians. Third, to the best of our knowledge, the present study is the first one assessing the predictive value, in the sense of patient temporary clinical deterioration or improvement, of the various factors and variables (coagulation system components, etc) measured on ICU admission, regardless from the final outcome. Indeed, many previous studies have examined the predictive value of most of these factors and variables in the sense of the final outcome, i.e., survival or death (e.g., [11,12,13,21,22,23,24,25,26,43,44,45,46,47,48,49,63,64,73]).

We acknowledge some limitations. First, we included patients from only one Hospital, thereby potentially minimizing generalizability. Second, three of the groups of the study, i.e., SIRS without infection, sepsis, and severe sepsis, included a rather small number of patients (*n* = 14–17), thus making solid conclusions uncertain in these groups. Third, we did not measure coagulation factor FXIII activity which is a plasma transglutaminase responsible for final clot stabilitywith a strong role in inflammation and sepsis [80].

## 5. Conclusions

In patients admitted to the ICU with a suspected diagnosis of infection, among the coagulation factors only FVII and FIX and the coagulation inhibitors, ATIII and PrC had substantial value in the sense of predicting temporary clinical improvement or deterioration in patients with severe sepsis or septic shock. Cytokine levels had a significant predictive value only in the group of patients with uncomplicated sepsis, while levels of CRP, PCT and TPO were predictive only in the group of patients with severe sepsis. Lactate had a significant predictive value only in the group of patients with septic shock, whereas the various clinico-laboratory scores and measurements (APACHE II, SOFA, LIS, and PaO_2_/FiO_2_) had a significant predictive value in almost all groups of septic patients.

## Figures and Tables

**Figure 1 jcm-10-01548-f001:**
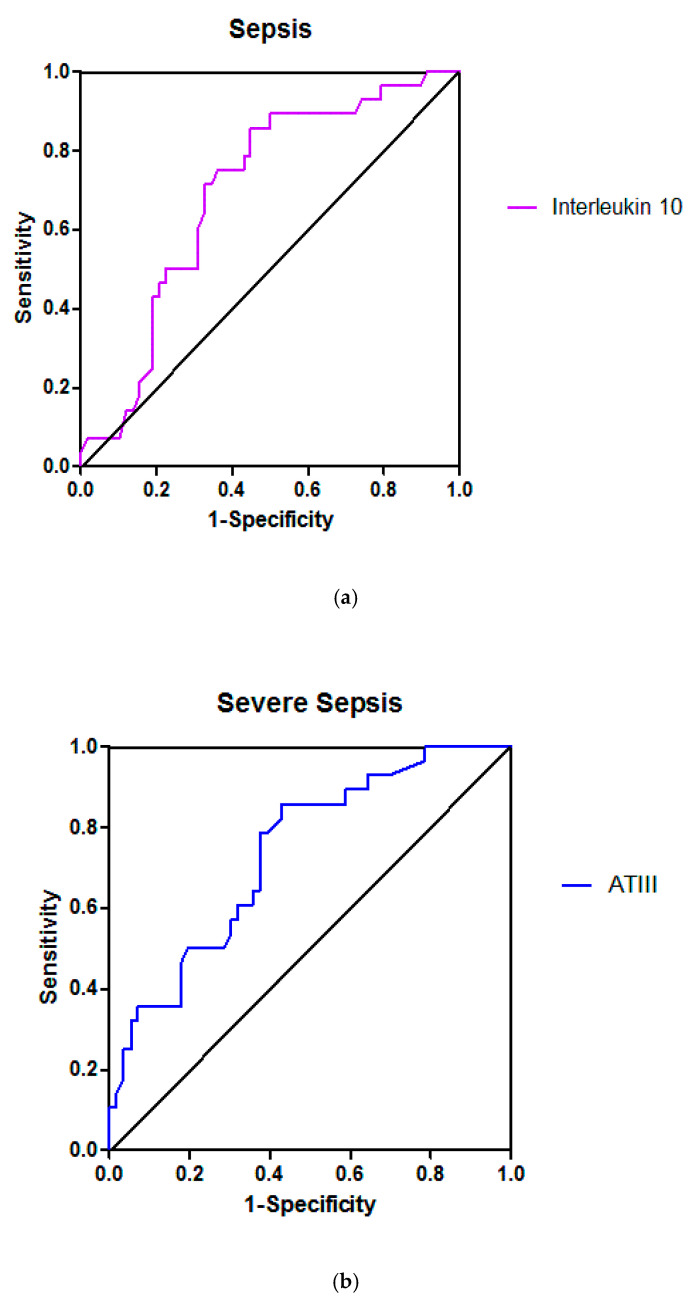
Receiver operating characteristic curves for Inteleukin 10 (**a**), Antithrombin III (ATIII) (**b**) and Protein C (**c**) in sepsis, severe sepsis and septic shock, respectively; the areas under the curves, indicating the predictive value of each variable in the sense of patient temporary clinical deterioration or improvement, were 0.72 ([95% confidenceinterval] 0.51–0.87), 0.74 (0.62−0.86) and 0.78 (0.60–0.95), respectively.

**Table 1 jcm-10-01548-t001:** Clinical and laboratory characteristics of patients on intensive care unit (ICU) admission.

	Τrauma–Surgery (*n* = 26)			SIRS (*n* = 17)			Sepsis (*n* = 14)			Severe Sepsis (*n* = 17)			Septic Shock (*n* = 28)			*p ^a^*
	Median	Range		Median	Range		Median	Range		Median	Range		Median	Range		
**Age** (yrs)	**38**	17	77	**53**	17	80	**43**	25	66	**55**	32	76	**49**	24	70	0.07
**APACHE**	**7.0**	2.0	18.0	**11.5** *^b^*	5.0	24.0	**13.0** *^b^*	5.0	21.0	**16.0** *^b,c^*	6.0	27.0	**23.0** *^b,c,d^*	8.0	31.0	**<0.001**
**SOFA**	**6.0**	2.0	14.0	**8.0**	1.0	13.0	**2.0** *^b,c^*	1.0	5.0	**8.0** *^b,d^*	6.0	11.0	**10.0** *^b,c,d^*	6.0	14.0	**<0.01**
**LIS**	**0.6**	0.0	2.0	**1.0**	0.0	3.3	**0.6**	0.0	2.0	**1.8** *^b,c,d^*	0.60	2.9	**1.9** *^b,c,d^*	0.2	3.9	**0.04**
**PaO_2_/FiO_2_** (mmHg)	**301**	90	647	**186**	110	587	**166** *^b^*	132	337	**112** *^b,c,d^*	66	326	**111** *^b,c,d^*	70	325	**<0.01**
**MODS** (yes/no)	3/23 (**12%**)			4/13 (**24%**)			0/14 (**0%**) *^b,c^*			7/10 (**41%**) *^b,c,d^*			16/12 (**57%**) *^b,c,d,e^*			
**Ht** (%)	**29.0**	19.6	43.4	**29.2**	19.0	41.8	**30.0**	24.0	35.8	**30.5**	24.3	35.0	**27.5**	15.0	36.5	0.71
**PLT** (×10^9^/L)	**195**	80	502	**238**	52	790	**233**	87	677	**164** *^c,d^*	33	430	**135** *^b,c,d,e^*	15	338	**<0.01**
**WBC** (×10^9^/L)	**9.1**	5.7	20.4	**10.6**	3.6	24.0	**13.4** *^b^*	6.3	33.0	**14.5** *^b^*	4.4	39.1	**17.3** *^b,c,d^*	6.4	41.3	**<0.01**
**Glucose** (mg/dL)	**136**	85	254	**140**	82	568	**115**	106	158	**135**	131	199	**150**	138	223	0.09
**Lactate** (mmol/L)	**1.3**	0.7	6.7	**2.0**	0.7	4.9	**1.9**	0.6	3.5	**1.6**	0.7	3.0	**3.7** *^b,c,d,e^*	0.8	8.9	**<0.01**
**Creatinin** (mg/dL)	**0.8**	0.5	2.7	**0.9**	0.4	4.9	**0.9**	0.6	4.1	**1.8** *^b,c,d^*	0.7	5.2	**1.9** *^b,c,d^*	0.4	9.0	**<0.01**
**Bil** (mg/dL)	**0.6**	0.2	2.4	**0.7**	0.1	2.6	**1.1**	0.4	5.2	**1.5** *^b,c,d^*	0.5	14.2	**1.6** *^b,c,d^*	0.2	18.9	**<0.01**
**Alb** (g/dL)	**2.8**	2.0	3.9	**2.6**	1.8	4.4	**2.5**	1.4	3.3	**2.7**	2.0	3.9	**2.1** *^b,c,d,e^*	1.2	3.7	**0.02**

ICU, intensive care unit; SIRS, Systemic Inflammatory Response Syndrome (4–7); APACHE, Acute Physiology and Chronic Health Evaluation (14); SOFA, Sequential Organ Failure Assessment (16); LIS; Lung Injury Score (17); PaO_2_, partial pressure of arterial oxygen; FiO_2_, fraction of inspired oxygen; MODS, Multiple Organ Dysfunction Syndrome (15); Ht, hematocrit; PTL, platelets; WBC, white blood cells; Bil, bilirubin; Alb, albumin. *^a^* corresponds to the comparison between all groups by the Kruskal Wallis test; *^b^* significantly different from trauma/surgery (*p* < 0.05); *^c^* significantly different from patients with SIRS (*p* < 0.05); *^d^* significantly different from patients with sepsis (*p* < 0.05); *^e^* significantly different from patients with severe sepsis (*p* < 0.05).

**Table 2 jcm-10-01548-t002:** Measurements of the coagulation system, cytokines, and inflammatory markers of patients on ICU admission.

	Τrauma–Surgery (*n* = 26)			SIRS (*n* = 17)			Sepsis (*n* = 14)			Severe Sepsis (*n* = 17)			Septic Shock (*n* = 28)			*p ^a^*
	Median	Range		Median	Range		Median	Range		Median	Range		Median	Range		
**PT** (sec)	**14.0**	12.2	21.6	**13.9**	12.4	22.0	**15.4**	12.8	19.7	**16.5** *^b,c^*	13.1	27.2	**17.8** *^b,c^*	13.6	25.7	**<0.01**
**a-PTT** (sec)	**36.8**	26.5	57.5	**36.0**	25.6	70.4	**34.5**	32.3	50.5	**38.6**	36.7	56.0	**39.9**	27.8	51.7	0.12
**Fibrinogen** (mg/dL)	**438**	170	753	**423**	159	639	**511**	497	526	**535**	493	636	**504**	439	700	0.26
**Dds** (μg/mL)	**3.4**	0.3	7.4	**2.4**	0.6	9.8	**2.9**	0.7	4.2	**6.1** *^b,c,d^*	1.4	23.8	**9.2** *^b,c,d^*	1.8	37.6	**<0.01**
**FVII** (%)	**58.0**	36.0	94.0	**75.0** *^b^*	45.0	1410	**86.4** *^b^*	44.8	137.7	**61.3** *^,c,d^*	33.0	110.0	**34.0** *^b,c,d,e^*	29.0	80.0	**<0.001**
**FIX** (%)	**86.0**	75.0	111.0	**103.0**	91.0	133.0	**87.5**	65.0	110.0	**90.0**	76.0	114.0	**98.5**	75.0	128.0	0.13
**FVIII** (%)	**117.0**	67.0	146.0	**123.0**	55.6	151.0	**121.0**	88.0	146.0	**110.0**	71.6	148.0	**114.0**	81.0	162.0	0.35
**vWF** (%)	**148.5**	134.0	200.0	**111.0** *^b^*	95.0	177.0	**114.5** *^b^*	81.0	173.0	**128.0**	82.0	158.0	**136.5** *^c,d^*	85.0	461.0	**<0.001**
**FX** (%)	**58.5**	32.0	83.0	**92.5** *^b^*	74.0	116.0	**91.0** *^b^*	82.0	128.0	**86.0** *^b^*	73.0	113.0	**79.5** *^b^*	66.0	119.0	**<0.01**
**FV** (%)	**73.0**	29.0	133.0	**97.0** *^b^*	55.0	136.0	**105.0** *^b^*	61.0	129.0	**95.0** *^b^*	61.0	117.0	**76.0** *^c,d,e^*	34.0	111.0	**<0.01**
**ATIII** (%)	**66.0**	33.0	104.0	**86.0** *^b^*	66.0	131.0	**75.0** *^b^*	55.0	129.0	**50.3** *^b,c,d^*	29.0	102.0	**46.0** *^b,c,d^*	23.0	113.0	**<0.001**
**PrC** (%)	**76.5**	33.0	120.0	**77.0**	49.0	119.6	**81.0**	57.0	141.0	**79.1**	34.0	149.5	**58.0** *^b,c,d,e^*	12.0	125.0	**<0.01**
**Plasminogen** (%)	**75.3**	50.0	97.0	**62.0**	32.0	96.0	**71.3**	46.8	101.4	**63.4**	45.0	93.8	**58.7**	46.3	109.0	0.08
**TNF-a** (pg/mL)	**1.2**	0.0	9.7	**2.8**	0.7	5.0	**2.9**	0.5	5.4	**3.1**	1.2	5.1	**4.9** *^b,c,d^*	1.0	8.4	**0.02**
**IL-1b** (pg/mL)	**0.2**	0.0	1.9	**0.2**	0.1	3.9	**0.2**	0.1	1.5	**1.9**	0.7	3.1	**2.9** *^b,c,d^*	0.2	10.2	**0.04**
**IL-10** (pg/mL)	**5.1**	1.2	450.7	**9.8** *^b^*	1.9	438.2	**10.9** *^b^*	2.9	123.5	**11.8** *^b^*	1.77	211.8	**12.1** *^b^*	2.1	285.9	048
**IL-6** (pg/mL)	**22.2**	8.6	70.9	**24.0**	17.5	127.4	**26.6**	20.1	112.0	**29.2**	19.2	120.2	**31.6**	16.7	128.6	0.22
**CRP** (mg/dL)	**12.3**	0.1	26.0	**8.9**	1.6	30.9	**17.3**	0.4	49.7	**22.1** *^b,c^*	10.1	50.3	**27.9** *^b,c,d^*	5.3	73.0	**<0.01**
**PCT** (ng/mL)	**1.0**	0.0	31.6	**0.7**	0.1	10.0	**0.5**	0.1	43.3	**5.7** *^b,c,d^*	0.2	130.7	**4.4** *^b,c,d^*	0.3	110.8	**<0.01**
**TPO** (pg/mL)	**147**	11	583	**132**	21	2570	**142**	38	1160	**492** *^b,c,d^*	40	1344	**530** *^b,c,d^*	42	2213	**<0.01**

ICU, intensive care unit; SIRS, Systemic Inflammatory Response Syndrome (4-7); PT, prothrombin time; a-PTT, activated partial thromboplastin time; Dds, D-dimmers; FVII, FIX, FVIII, FX, FV: coagulation factors VII, IX, VIII, X, and V, respectively; vWF, coagulation factor vWillebrand; ATIII, Antithrombin III; PrC, Protein C; TNF-a, Tumor Necrosis Factor a; IL-1b, Interleukin 1β; IL-10, Interleukin 10; IL-6, Interleukin 6; CRP, C-reactive protein; PCT, Procalcitonin; TPO, Thrombopoietin. *^a^* corresponds to the comparison between all groups by the Kruskal–Wallis test; *^b^* significantly different from trauma/surgery (*p* < 0.05); *^c^* significantly different from patients with SIRS (*p* < 0.05); *^d^* significantly different from patients with sepsis (*p* < 0.05); *^e^* significantly different from patients with severe sepsis (*p* < 0.05).

**Table 3 jcm-10-01548-t003:** Diagnosis of trauma or surgery without suspected infection on ICU admission.

	Patients Who Deteriorated (*n* = 12)			Patients Who Improved (*n* = 14)			*p ^a^*
	Median	Range		Median	Range		
**Age** (yrs)	**32**	17	77	**50**	22	77	0.16
**APACHE**	**8.0**	4.0	16.5	**7.0**	1.5	18.0	0.15
**SOFA**	**7.0**	4.0	14.0	**5.5**	2.0	7.0	0.15
**LIS**	**1.0**	0.0	2.0	**0.3**	0.0	2.0	0.09
**PaO_2_/FiO_2_** (mmHg)	**201**	90	542	**350**	148	647	**0.04**
**MODS** (yes/no)	2/10 (**17%**)			1/13 (**7%**)			0.58 *^b^*
**Ht** (%)	**27.7**	19.6	39.9	**31.0**	28.2	43.4	0.11
**PLT** (×10^9^/L)	**131**	80	502	**226**	120	337	0.09
**WBC** (×10^9^/L)	**8.6**	5.7	20.4	**12.3**	7.9	19.6	0.08
**Glu** (mg/dL)	**128**	111	221	**140**	85	254	0.95
**Lactate** (mmol/L)	**1.5**	0.9	6.7	**1.2**	0.7	2.9	0.74
**Creatin** (mg/dL)	**0.7**	0.5	2.7	**0.9**	0.6	1.8	0.28
**Bil** (mg/dL)	**0.7**	0.2	2.3	**0.5**	0.4	2.4	0.90
**Alb** (g/dL)	**3.0**	2.2	3.9	**2.5**	2.0	3.6	0.92
**PT** (sec)	**15.2**	12.9	21.6	**12.6**	12.2	15.5	**<0.001**
**a-PTT** (sec)	**39.8**	28.00	57.5	**33.6**	26.5	44.4	0.39
**Fibrin** (mg/dL)	**351**	170	753	**457**	322	622	0.52
**Dds** (μg/mL)	**4.6**	0.6	7.4	**3.0**	0.3	3.2	**0.04**
**FVII** (%)	**46.5**	26.00	94.0	**65.0**	35	94	0.25
**FIX** (%)	**84.0**	75.00	107.0	**93.0**	79.0	111.0	0.33
**FVIII** (%)	**101.5**	67.00	146.0	**120.0**	73.0	140.0	**0.04**
**vWF** (%)	**150.0**	142.5	200.0	**147.0**	134.0	195.0	0.84
**FX** (%)	**51.2**	32.00	83.0	**60.0**	50.00	76.0	0.54
**FV** (%)	**66.0**	29.00	130.0	**81.0**	43.00	133.0	0.28
**ATIII** (%)	**64.0**	33.00	104.0	**72.9**	51.0	99.0	0.90
**PrC** (%)	**72.0**	33.0	115.0	**77.5**	65.00	120.0	0.60
**Plasminogen** (%)	**59.0**	50.0	97.0	**78.0**	63.1	83.0	0.12
**TNF-a** (pg/mL)	**2.1**	1.3	9.7	**0.9**	0.00	1.3	**<0.001**
**IL-1b** (pg/mL)	**0.2**	0.1	1.8	**0.2**	0.00	1.9	0.94
**IL-10** (pg/mL)	**6.7**	1.6	356.3	**4.8**	1.2	450.7	0.49
**IL-6** (pg/mL)	**23.2**	12.0	53.0	**21.0**	8.6	70.9	0.41
**CRP** (mg/dL)	**12.8**	2.3	22.1	**11.9**	0.1	26.0	0.69
**PCT** (ng/mL)	**1.1**	0.21	24.5	**0.6**	0.0	31.6	0.65
**TPO** (pg/mL)	**177**	33	583	**127**	11	367	0.60

For explanation of abbreviations, see footnotes of Table 1 and Table 2. *^a^* corresponds to the comparison between the two groups by the Mann–Whitney test; *^b^* corresponds to the comparison between the two groups by the Fisher’s exact test.

**Table 4 jcm-10-01548-t004:** Diagnosis of Systemic Inflammatory Response Syndromeon ICU admission.

	Patients Who Deteriorated (*n* = 13)			Patients Who Improved (*n* = 4)			*p ^a^*
	Median	Range		Median	Range		
**Age** (yrs)	**48**	17	80	**56**	43	69	0.35
**APACHE**	**11.5**	5.0	24.0	**14.5**	8.0	22.0	0.62
**SOFA**	**8.0**	1.0	66.0	**9.5**	6.0	10.0	0.62
**LIS**	**1.3**	0.0	3.3	**0.6**	0.3	2.0	0.52
**PaO_2_/FiO_2_** (mmHg)	**164**	46	587	**226**	136	280	**0.04**
**MODS** (yes/no)	4/9 (**31%**)			0/4 (**0%**)			0.51 *^b^*
**Ht** (%)	**29.2**	19.0	41.5	**36.7**	28.9	41.8	0.13
**PLT** (×10^9^/L)	**152**	52	279	**179**	140	790	0.24
**WBC** (×10^9^/L)	**10.6**	3.6	16.0	**10.9**	9.0	24.0	0.62
**Glu** (mg/dL)	**140**	108	568	**156**	82	311	1.00
**Lactate** (mmol/L)	**2.0**	0.7	4.9	**2.0**	1.3	2.2	0.86
**Creatin** (mg/dL)	**0.9**	0.4	4.9	**0.9**	0.7	1.2	0.70
**Bil** (mg/dL)	**0.7**	0.3	2.6	**0.8**	0.1	1.4	0.87
**Alb** (g/dL)	**2.5**	1.8	4.4	**2.8**	1.8	3.9	0.75
**PT** (sec)	**13.9**	12.4	22.0	**14.6**	13.1	16.4	0.78
**a-PTT** (sec)	**36.6**	25.6	70.4	**34.4**	30.3	38.5	0.41
**Fibrin** (mg/dL)	**478**	159	639	**336**	292	405	0.10
**Dds** (μg/mL)	**3.1**	0.6	9.8	**1.6**	0.8	2.4	0.66
**FVII** (%)	**63.5**	45.0	119.0	**78.0**	75.0	141.0	0.28
**FIX** (%)	**103.0**	92.0	133.0	**105.0**	91.0	119.0	0.80
**FVIII** (%)	**122.5**	55.6	140.0	**126.0**	81.0	151.0	0.57
**vWF** (%)	**104.0**	95.0	177.0	**127.0**	99.0	150.0	0.20
**FX** (%)	**86.5**	74.0	105.0	**102.5**	89.0	116.0	0.53
**FV** (%)	**65.0**	55.0	136.0	**110.0**	79.0	126.0	0.07
**ATIII** (%)	**74.0**	66.0	86.0	**102.0**	85.0	131.0	0.57
**PrC** (%)	**73.5**	49.0	119.6	**87.0**	52.0	108.5	0.69
**Plasminogen** (%)	**66.5**	32.0	96.0	**59.4**	44.7	91.0	0.67
**TNF-a** (pg/mL)	**2.9**	0.7	5.0	**2.6**	0.8	4.9	1.00
**IL-1b** (pg/mL)	**0.4**	0.1	3.9	**0.2**	0.2	3.4	0.69
**IL-10** (pg/mL)	**8.6**	1.9	80.0	**58.8**	9.8	438.2	0.13
**IL-6** (pg/mL)	**22.0**	17.5	127.4	**26.6**	13.7	113.8	0.81
**CRP** (mg/dL)	**8.3**	1.6	30.9	**9.9**	2.4	17.4	0.60
**PCT** (ng/mL)	**1.0**	0.10	10.0	**0.4**	0.3	0.6	0.52
**TPO** (pg/mL)	**95.0**	21	2570	**338**	126	941	0.22

For explanation of abbreviations, see footnotes of Table 1 and Table 2. *^a^* corresponds to the comparison between the two groups by the Mann–Whitney test; *^b^* corresponds to the comparison between the two groups by the Fisher’s exact test.

**Table 5 jcm-10-01548-t005:** Diagnosis of Sepsis on ICU admission.

	Patients Who Deteriorated (*n* = 10)			Patients Who Improved (*n* = 4)			*p ^a^*
	Median	Range		Median	Range		
**Age** (yrs)	**45**	29	58	**35**	25	66	0.46
**APACHE**	**14.0**	6.0	21.0	**7.6**	5.0	10.0	**<0.01**
**SOFA**	**3.5**	2.0	5.0	**1.5**	1.0	2.0	**0.01**
**LIS**	**1.5**	0.5	2.0	**0.4**	0.0	1.8	**0.03**
**PaO_2_/FiO_2_** (mmHg)	**183**	132	211	**315**	154	337	0.08
**MODS** (yes/no)	0/10 (**0%**)			0/4 (**0%**)			1.00 *^b^*
**Ht** (%)	**27.9**	24.0	35.8	**32.1**	25.0	34.2	0.32
**PLT** (×10^9^/L)	**195**	87	462	**267**	194	677	0.09
**WBC** (×10^9^/L)	**12.5**	6.3	21.0	**14.7**	8.3	33.0	0.83
**Glu** (mg/dL)	**122**	106	158	**112**	108	126	0.65
**Lactate** (mmol/L)	**1.4**	0.9	3.5	**2.1**	0.6	2.6	0.21
**Creatin** (mg/dL)	**1.0**	0.6	4.1	**0.8**	0.7	1.4	0.08
**Bil** (mg/dL)	**1.2**	0.4	5.2	**1.0**	0.5	2.5	0.34
**Alb** (g/dL)	**2.3**	1.4	3.3	**2.8**	2.5	3.3	0.26
**PT** (sec)	**15.6**	13.5	19.7	**14.5**	12.8	17.9	0.14
**a-PTT ** (sec)	**38.6**	33.6	45.4	**34.0**	32.3	50.5	0.07
**Fibrin** (mg/dL)	**516**	497	518	**498**	480	526	0.97
**Dds** (μg/mL)	**2.7**	1.4	4.2	**3.0**	0.7	4.0	0.84
**FVII** (%)	**89.0**	44.8	97.0	**101.0**	73.0	137.7	0.23
**FIX** (%)	**84.7**	75.0	106.0	**96.0**	65.0	110.0	0.37
**FVIII** (%)	**132.3**	90.3	146.0	**102.7**	88.0	134.5	0.79
**vWF** (%)	**108.0**	81.0	170.3	**139.0**	124.0	173.0	0.16
**FX** (%)	**89.0**	82.0	125.0	**95.5**	89.0	128.0	0.58
**FV** (%)	**89.5**	67.8	108.9	**115.8**	61.0	129.0	0.41
**ATIII** (%)	**78.8**	55.0	120.5	**83.1**	57.0	141.0	0.24
**PrC** (%)	**68.3**	5.0	138.2	**85.0**	37.0	168.4	0.40
**Plasminogen** (%)	**78.8**	71.7	100.1	**64.4**	46.8	101.4	0.47
**TNF-a** (pg/mL)	**4.9**	0.8	5.4	**1.1**	0.5	2.3	**0.02**
**IL-1b** (pg/mL)	**1.2**	0.4	1.5	**0.3**	0.1	0.7	**<0.001**
**IL-10** (pg/mL)	**17.4**	3.3	123.5	**3.4**	2.9	10.0	**<0.001**
**IL-6** (pg/mL)	**24.0**	11.0	26.4	**31.5**	20.1	112.0	0.82
**CRP** (mg/dL)	**19.6**	5.1	49.7	**11.9**	0.4	16.9	0.08
**PCT** (ng/mL)	**1.1**	0.1	39.0	**0.3**	0.1	43.3	0.44
**TPO** (pg/mL)	**168**	23	1160	**124**	38	397	0.25

For explanation of abbreviations, see footnotes of Table 1 and Table 2. *^a^* corresponds to the comparison between the two groups by the Mann–Whitney test; *^b^* corresponds to the comparison between the two groups by the Fisher’s exact test.

**Table 6 jcm-10-01548-t006:** Diagnosis of Severe Sepsis on ICU admission.

	Patients Who Deteriorated (*n* = 9)			Patients Who Improved (*n* = 8)			*p ^a^*
	Median	Range		Median	Range		
**Age (yrs)**	**70**	32	76	**40**	33	74	**0.03**
**APACHE**	**19.0**	11.0	27.0	**13.0**	6.0	21.0	**<0.01**
**SOFA**	**9.5**	7.0	11.0	**7.5**	6.0	11.0	**0.04**
**LIS**	**2.6**	0.6	2.9	**1.3**	0.6	2.0	**0.03**
**PaO_2_/FiO_2_** (mmHg)	**96**	66	326	**281**	136	320	**<0.01**
**MODS** (yes/no)	5/4(**56%**)			2/6 (**25%**)			0.33 *^b^*
**Ht** (%)	**29.5**	24.3	34.5	**32.7**	25.00	35.0	0.67
**PLT** (×10^9^/L)	**140**	33	230	**187**	93	430	0.54
**WBC** (×10^9^/L)	**12.4**	6.9	19.9	**16.3**	4.4	39.1	0.96
**Glu** (mg/dL)	**133**	131	197	**149**	135	199	1.00
**Lactate** (mmol/L)	**2.5**	1.6	3.0	**1.1**	0.7	2.0	0.07
**Creatin** (mg/dL)	**2.3**	0.7	5.2	**1.7**	0.7	2.0	0.08
**Bil** (mg/dL)	**3.5**	0.6	14.2	**0.8**	0.5	3.5	0.06
**Alb** (g/dL)	**2.4**	2.0	2.9	**2.9**	1.80	3.9	0.87
**PT** (sec)	**17.5**	13.6	27.2	**14.3**	13.1	19.7	0.22
**a-PTT** (sec)	**42.3**	36.7	56.0	**35.0**	37.8	51.7	0.23
**Fibrin** (mg/dL)	**514**	493	600	**542**	499	636	1.00
**Dds** (μg/mL)	**7.8**	1.4	23.8	**4.5**	1.8	6.2	0.65
**FVII** (%)	**41.0**	33.0	65.0	**66.0**	45.0	110.0	0.06
**FIX** (%)	**84.0**	76.0	104.0	**107.5**	96.0	114.0	0.09
**FVIII** (%)	**104.0**	71.6	145.0	**122.0**	81.6	148.0	0.39
**vWF** (%)	**99.0**	82.0	158.0	**130.5**	114.00	134.0	0.67
**FX** (%)	**79.0**	73.0	112.0	**99.5**	90.0	113.0	0.62
**FV** (%)	**85.0**	61.0	117.0	**101.0**	81.0	115.0	0.98
**ATIII** (%)	**33.2**	29.0	70.0	**66.0**	40.3	102.0	**0.02**
**PrC** (%)	**44.0**	34.0	49.0	**97.0**	58.0	149.5	**<0.001**
**Plasminogen** (%)	**56.3**	45.0	93.0	**73.4**	67.3	93.8	0.32
**TNF-a** (pg/mL)	**4.1**	3.2	5.1	**2.6**	1.2	4.4	0.18
**IL-1b** (pg/mL)	**1.6**	0.9	3.1	**0.9**	0.7	1.7	0.45
**IL-10** (pg/mL)	**15.4**	2.1	211.8	**5.4**	1.8	85.9	0.02
**IL-6** (pg/mL)	**26.0**	20.2	120.2	**31.7**	19.2	43.6	0.85
**CRP** (mg/dL)	**32.4**	21.6	50.3	**10.1**	10.1	17.9	**<0.01**
**PCT** (ng/mL)	**8.5**	1.4	130,7	**0.7**	0.2	1.5	**<0.01**
**TPO** (pg/mL)	**644**	194	1344	**160.0**	40	519	**<0.01**

For explanation of abbreviations, see footnotes of Table 1 and Table 2. *^a^* corresponds to the comparison between the two groups by the Mann–Whitney test; *^b^* corresponds to the comparison between the two groups by the Fisher’s exact test.

**Table 7 jcm-10-01548-t007:** Diagnosis of Septic Shock on ICU admission.

	Patients Who Deterioratedand Died (*n* = 17)			Patients Who Improved (*n* = 11)			*p ^a^*
	Median	Range		Median	Range		
**Age** (yrs)	**50**	27	70	**36**	24	70	0.45
**APACHE**	**25.6**	12.5	31.0	**15.4**	8.0	21.0	**<0.01**
**SOFA**	**11.5**	7.0	14.0	**8.8**	6.0	14.0	**0.03**
**LIS**	**2.5**	0.2	3.9	**1.1**	0.2	2.0	**<0.01**
**PaO_2_/FiO_2_** (mmHg)	**103**	70	320	**248**	136	325	**<0.01**
**MODS** (yes/no)	15/2 (**88%**)			3/8(**27%**)			**<0.01 *^b^***
**Ht** (%)	**26.4**	19.3	31.0	**28.5**	15.0	36.5	0.59
**PLT** (×10^9^/L)	**122**	15	336	**206**	93	338	0.06
**WBC** (×10^9^/L)	**18.6**	6.4	41.3	**13.8**	11.0	31.3	0.69
**Glu** (mg/dL)	**160**	138	223	**133**	104	222	0.33
**Lactate** (mmol/L)	**4.4**	1.2	8.9	**1.7**	0.8	2.4	**<0.001**
**Creatin** (mg/dL)	**2.3**	0.5	9.0	**1.2**	0.4	2.2	0.13
**Bil** (mg/dL)	**3.1**	0.5	18.9	**0.9**	0.2	3.1	0.08
**Alb** (g/dL)	**2.0**	1.2	3.5	**2.2**	2.3	3.7	0.81
**PT** (sec)	**19.9**	13.3	25.7	**16.8**	13.6	19.7	0.23
**a-PTT** (sec)	**55.1**	35.1	180.0	**36.7**	27.8	51.7	0.12
**Fibrin** (mg/dL)	**458**	439	700	**644**	444	699	0.24
**Dds** (μg/mL)	**10.8**	1.8	37.6	**4.1**	1.9	14.6	0.06
**FVII** (%)	**30.5**	29.0	61.0	**58.3**	40.0	80.0	**<0.001**
**FIX** (%)	**82.2**	75.0	112.0	**101.0**	95.0	128.0	**0.03**
**FVIII** (%)	**108.4**	81.0	158.3	**119.8**	104.1	162.0	0.81
**vWF**(%)	**123.7**	85.0	252.0	**146.0**	114.0	461.0	0.23
**FX** (%)	**73.4**	66.0	117.5	**90.0**	75.00	119.0	0.12
**FV** (%)	**69.7**	34.0	109.5	**89.9**	48.0	111.0	0.37
**ATIII** (%)	**30.7**	23.0	83.0	**57.6**	38.0	113.0	0.21
**PrC** (%)	**4** **8.** **0**	12.0	68.0	**63.** **0**	47.5	125.0	**0.01**
**Plasminogen** (%)	**52.3**	46.3	109.0	**64.0**	90.50	108.4	0.40
**TNF-a** (pg/mL)	**5.1**	1.0	8.4	**4.5**	1.0	8.2	0.85
**IL-1b** (pg/mL)	**3.1**	0.3	9.4	**1.9**	0.2	10.2	0.33
**IL-10** (pg/mL)	**11.6**	2.1	285.9	**15.8**	2.5	160.8	0.23
**IL-6** (pg/mL)	**33.0**	16.7	128.6	**30.9**	15.4	86.0	0.64
**CRP** (mg/dL)	**25.6**	5.3	72.7	**31.2**	5.27	73.0	0.62
**PCT** (ng/mL)	**4.7**	0.4	110.8	**3.8**	0.3	16.5	0.27
**TPO** (pg/mL)	**601**	47	2213	**487**	42	774	0.18

For explanation of abbreviations, see footnotes of Table 1 and Table 2. *^a^* corresponds to the comparisn between the two groups by the Mann–Whitney test; *^b^* corresponds to the comparison between the two groups by the Fisher’s exact test.

**Table 8 jcm-10-01548-t008:** Predictive value (Area Under the Receiver Operating Characteristic—AUROC (95% CI)), in the sense of patient temporary clinical deterioration or improvement, of the various factors and variables measured on ICU admission.

	Τrauma–Surgery	SIRS	Sepsis	Severe Sepsis	Septic Shock
**Age** (yrs)				**0.66** (0.57–0.76)	
**APACHE**			**0.84** (0.73–0.96)	**0.76** (0.65–0.88)	**0.83** (0.67–0.98)
**SOFA**			**0.82** (0.65–0.99)	**0.73** (0.56–0.90)	**0.72** (0.55–0.89)
**LIS**			**0.82** (0.66–1.00)	**0.80** (0.63–0.98)	**0.78** (0.65–0.90)
**PaO_2_/FiO_2_** (mmHg)	**0.67** (0.56–0.78)	**0.64** (0.44–0.83)		**0.90** (0.80–0.99)	**0.79** (0.66–0.92)
**MODS** (yes/no)					**0.91** (0.82–1.00)
**Ht** (%)					
**PLT** (×10^9^/L)					
**WBC** (×10^9^/L)					
**Glu** (mg/dL)					
**Lactate** (mmol/L)					**0.87** (0.79–0.94)
**Creatin** (mg/dL)					
**Bil** (mg/dL)					
**Alb** (g/dL)					
**PT** (sec)	**0.75** (0.58–0.83)				
**a-PTT** (sec)					
**Fibrin** (mg/dL)					
**Dds** (μg/mL)	**0.71** (0.64–0.79)				
**FVII** (%)					**0.72** (0.49–0.83)
**FIX** (%)					**0.67** (0.53-0.78)
**FVIII** (%)	**0.63** (0.49–0.76)				
**vWF**(%)					
**FX** (%)					
**FV** (%)					
**ATIII** (%)				**0.74** (0.62–0.86)	
**PrC** (%)				**0.75** (0.57–0.83)	**0.78** (0.60–0.95)
**Plasminogen** (%)					
**TNF-a** (pg/mL)	**0.76** (0.65–0.89)		**0.66** (0.53–0.76)		
**IL-1b** (pg/mL)			**0.71** (0.48–0.83)		
**IL-10** (pg/mL)			**0.72** (0.51–0.87)		
**IL-6** (pg/mL)					
**CRP** (mg/dL)				**0.74** (0.60–0.89)	
**PCT** (ng/mL)				**0.75** (0.56–0.89)	
**TPO** (pg/mL)				**0.73** (0.46–0.91)	

AUROC, area under the receiver operating characteristic curve; CI, confidence interval; SIRS, systemic inflammatory response syndrome. For explanation of the other abbreviations, see footnotes of Table 1 and Table 2.

## Data Availability

The datasets used/or analyzed in the present study are available from the corresponding author on reasonable request.

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
