# Peer review of "Can Coagulation System Disorders and Cytokine and Inflammatory Marker Levels Predict the Temporary Clinical Deterioration or Improvement of Septic Patients on ICU Admission?"

_jcm, 2021, doi:10.3390/jcm10081548_

Round 1
Reviewer 1 Report
In this manuscript, Lavranou and co-workers examine whether various serum coagulation and cyto/chemokine biomarkers are valuable predictors of clinical outcomes in critically ill patients. Although many biomarker studies have been published in this patient population, thus limiting the novelty somewhat, this study is strengthened by many factors, including careful classification of patients into 5 groups- sepsis, severe sepsis, septic 102 shock, SIRS without infection, and trauma or surgery without SIRS or infection. The endpoints are well-described and appropriately presented. Overall the study seems to be very well done and the results are interesting.
One important characteristic that has not been included in this analysis is obesity, or the presence of underlying cardiometabolic disorders. Indeed, it is somewhat surprising that the authors have not included, at the very least, BMI in their predictive analysis, given that obese individuals are known to have a more severe inflammatory response and a poor survival in sepsis compared with normal weight individuals. Moreover, given that coagulation endpoints are evaluated here, there needs to be some accounting for the presence of underlying coronary disease in the older patients, particularly since age was an important predictor of poor outcome (as would be expected).
Author Response
Comments of Reviewer #1
In this manuscript, Lavranou and co-workers examine whether various serum coagulation and cyto/chemokine biomarkers are valuable predictors of clinical outcomes in critically ill patients. Although many biomarker studies have been published in this patient population, thus limiting the novelty somewhat, this study is strengthened by many factors, including careful classification of patients into 5 groups- sepsis, severe sepsis, septic 102 shock, SIRS without infection, and trauma or surgery without SIRS or infection. The endpoints are well-described and appropriately presented. Overall the study seems to be very well done and the results are interesting.
R1: The authors would like to thank you for the time and effort taken to review our manuscript. We also thank you for your positive comments and critique of our manuscript. Addressing these points has greatly improved the quality of our manuscript and as such we are thankful to you.
One important characteristic that has not been included in this analysis is obesity, or the presence of underlying cardiometabolic disorders. Indeed, it is somewhat surprising that the authors have not included, at the very least, BMI in their predictive analysis, given that obese individuals are known to have a more severe inflammatory response and a poor survival in sepsis compared with normal weight individuals. Moreover, given that coagulation endpoints are evaluated here, there needs to be some accounting for the presence of underlying coronary disease in the older patients, particularly since age was an important predictor of poor outcome (as would be expected).
R2: Thank you for this comment. In the revised version of our manuscript we have added the information about comorbidities in two points of the Results section. First, in the first paragraph of the Results (3.1.), which reads as follows: «Some of the comorbidities (e.g., chronic lung disease) are incorporated in the chronic health point system of the APACHE II score. In more detail, 28/102 patients (27%) had hypertension, 14/102 (14%) had diabetes mellitus, 7/102 (7%) had coronary artery disease, 8/102 (8%) had congestive heart failure, 10/102 (10%) had obesity, defined as Body Mass Index > 30 kg/m2, and 3/102 patients (3%) had chronic lung disease, whereas 39/102 patients (38%) had at least two comorbidities.». Second, in the third paragraph of the Results (3.3.), which reads as follows: «Moreover, the percentages of the various comorbidities were not significantly different between patients who deteriorated or improved in all five patient groups.».
Again, thank you very much for your very careful review of our manuscript.
Reviewer 2 Report
The degree of coagulation disorders and inflammatory responses in septic patients, evaluated upon admission to ICU, could predict the outcome of sepsis. In the present study, 102 patients were divided into 5 groups, according to the severity of sepsis. Factors VII and IX, protein C, antithrombin III, C-reactive protein, procalcitonin, thrombopoietin, tumor necrosis factor a, and interleukins 1β and 10 were measured in systemic blood within the first 24 h of admission to ICU. To assess the predictive value of these factors, the AUROC approach was used. It is concluded that coagulation factors and inhibitors as well as cytokine and inflammatory marker levels have substantial predictive value in distinct groups of septic patients.
Despite many pre-clinical studies and clinical trials, there is no known treatment of sepsis. Based on assessment of the various factors at admission, does the present study aim to unravel the factors most critical for the outcome of sepsis, and thus point to the type of intervention needed, or to the direction of scientific/clinical research, to eventually improve the outcome? Please briefly address this question in the manuscript.
Comments.
(1) Lines 87-100: The measures described in this section are clinical measures used in the ICU. However, these measures have scientific rationale based on the current mechanistic understanding of sepsis. Please briefly describe in this section the scientific rationale why these measures were taken.
(2) Lines 117-119: Daily clinical and laboratory evaluation of the patients was performed until their discharge from the ICU or death. There were 5 groups of patients in the study, but there is no statement of how these groups were treated during their stay in ICU. There is also no statement of co-morbidities of these patients. Unless patients in all groups were treated the same way and they had no co-morbidities, it is difficult to determine whether the coagulation/cytokine/inflammatory markers actually predicted the deterioration or improvement of patients. This deterioration or improvement could have been due to differences in day-to-day treatment and/or co-morbidities of the patients. Please briefly describe in the paper the treatment of patients in the 5 groups and their co-morbidities.
(3) Line 173: Please explain in more detail the AUROC analysis. This could be done in the on-line supplemental material submitted with the paper. For example, for the protein C graph in Figure 1, show step-by-step how raw data for protein C levels obtained from septic shock patients were used to generate the graph in Figure 1.
(4) Results. The presentation of results is too long. Data are presented twice, in the tables and also in the text. Please reduce the length of the text.
(5) Discussion. It is too long. It includes some of the scientific rationale behind the clinical measures used in the present study. Please remove this rationale from Discussion and add it to Methods where these measures are first introduced.
Use Discussion to briefly interpret the meaning of your data. For example, your measurements of the coagulation system and the platelet count are based on systemic blood samples. Yet, the coagulation system dysfunction in sepsis could manifest itself mainly in the smallest blood vessels where microthrombi block the perfusion of these vessels, leading to organ dysfunction (DeBacker et al., 2002). The microthrombi formation is the result of sepsis-induced blood hypercoagulability in the microcirculation. Remarkably, hypocoagulability has been reported for systemic blood samples in septic patients (Johansson et al., 2010). One explanation of this apparent contradiction could be that the coagulation factors and activated/adhering platelets in sepsis are trapped in the microcirculation, and thus not available for assessment in systemic blood.
(6) Line 642: “… the present study is the first one assessing the predictive value, in the sense of patient temporary clinical deterioration or improvement …”. The following statement “Indeed, many previous studies have examined the predictive value of most of these factors and variables in the sense of final outcome …” appears to contradict the previous statement. In what aspect was your study the first to assess the predictive value?
(7) Typographic errors: lines 15-16 (“thetempoary”), 40 (“organ dysfunction” rather than “organic dysfunction”), 45, 51, 140, 236, 268, 698. There are problems with grammar on lines 53, 72-75, 79, 133-137, making the text difficult to follow.
Author Response
Comments of Reviewer #2
The degree of coagulation disorders and inflammatory responses in septic patients, evaluated upon admission to ICU, could predict the outcome of sepsis. In the present study, 102 patients were divided into 5 groups, according to the severity of sepsis. Factors VII and IX, protein C, antithrombin III, C-reactive protein, procalcitonin, thrombopoietin, tumor necrosis factor a, and interleukins 1β and 10 were measured in systemic blood within the first 24 h of admission to ICU. To assess the predictive value of these factors, the AUROC approach was used. It is concluded that coagulation factors and inhibitors as well as cytokine and inflammatory marker levels have substantial predictive value in distinct groups of septic patients.
R1: First of all, the authors would like to thank you for the time and effort taken to review our manuscript. We also thank you for your comments and critique in many parts of our manuscript. Addressing these points has greatly improved the quality of our manuscript and as such we are thankful to you.
Despite many pre-clinical studies and clinical trials, there is no known treatment of sepsis. Based on assessment of the various factors at admission, does the present study aim to unravel the factors most critical for the outcome of sepsis, and thus point to the type of intervention needed, or to the direction of scientific/clinical research, to eventually improve the outcome? Please briefly address this question in the manuscript.
R2: Thank you for this important comment. In the revised version of our manuscript we have incorporated the following statement in the beginning of the Discussion: «Despite many pre-clinical studies and clinical trials, there is no known treatment of sepsis. Based on assessment of the various factors at admission, the present study aimed to unravel the most critical factors for the outcome of sepsis, and thus point to the type of intervention needed, or to the direction of scientific/clinical research to eventually improve the outcome.».
Comments. (1) Lines 87-100: The measures described in this section are clinical measures used in the ICU. However, these measures have scientific rationale based on the current mechanistic understanding of sepsis. Please briefly describe in this section the scientific rationale why these measures were taken.
R3: In the revised version of our manuscript we have added this information in the Materials and Methods (at the end of the third paragraph of 2.1.), which reads as follows: «All these clinical measures have scientific rationale based on the current mechanistic understanding of sepsis and were taken by the attending physicians according to the guidelines used in our ICU [6].».
(2) Lines 117-119: Daily clinical and laboratory evaluation of the patients was performed until their discharge from the ICU or death. There were 5 groups of patients in the study, but there is no statement of how these groups were treated during their stay in ICU. There is also no statement of co-morbidities of these patients. Unless patients in all groups were treated the same way and they had no co-morbidities, it is difficult to determine whether the coagulation/cytokine/inflammatory markers actually predicted the deterioration or improvement of patients. This deterioration or improvement could have been due to differences in day-to-day treatment and/or co-morbidities of the patients. Please briefly describe in the paper the treatment of patients in the 5 groups and their co-morbidities
R4: Thank you for this comment. In the revised version of our manuscript we have added the information about treatment and comorbidities in two points of the Results section. First, in the first paragraph of the Results (3.1.), which reads as follows: «The treatment of patients was decided by the attending physicians according to the guidelines used in our ICU [6], and included administration of antibiotics, appropriate fluid management, etc. Some of the comorbidities (e.g., chronic lung disease) are incorporated in the chronic health point system of the APACHE II score. In more detail, 28/102 patients (27%) had hypertension, 14/102 (14%) had diabetes mellitus, 7/102 (7%) had coronary artery disease, 8/102 (8%) had congestive heart failure, 10/102 (10%) had obesity, defined as Body Mass Index > 30 kg/m2, and 3/102 patients (3%) had chronic lung disease, whereas 39/102 patients (38%) had at least two comorbidities.». Second, in the third paragraph of the Results (3.3.), which reads as follows: «The treatment was not different between patients who deteriorated or improved in all five patient groups. Moreover, the percentages of the various comorbidities were not significantly different between patients who deteriorated or improved in all five patient groups.».
(3) Line 173: Please explain in more detail the AUROC analysis. This could be done in the on-line supplemental material submitted with the paper. For example, for the protein C graph in Figure 1, show step-by-step how raw data for protein C levels obtained from septic shock patients were used to generate the graph in Figure 1.
R5: We now submit in the on-line supplemental material the following two outputs from SPSS: 1) Protein C descriptive statistics [the file is named: Protein C SPSS (descriptive statistics)], and 2) Protein C ROC curve production [the file is named: Protein C SPSS (ROC curve production)].
(4) Results. The presentation of results is too long. Data are presented twice, in the tables and also in the text. Please reduce the length of the text.
R6: We have tried to reduce the length of the text. Please see at the section 3.2. of the Results.
(5) Discussion. It is too long. It includes some of the scientific rationale behind the clinical measures used in the present study. Please remove this rationale from Discussion and add it to Methods where these measures are first introduced.
Use Discussion to briefly interpret the meaning of your data. For example, your measurements of the coagulation system and the platelet count are based on systemic blood samples. Yet, the coagulation system dysfunction in sepsis could manifest itself mainly in the smallest blood vessels where microthrombi block the perfusion of these vessels, leading to organ dysfunction (DeBacker et al., 2002). The microthrombi formation is the result of sepsis-induced blood hypercoagulability in the microcirculation. Remarkably, hypocoagulability has been reported for systemic blood samples in septic patients (Johansson et al., 2010). One explanation of this apparent contradiction could be that the coagulation factors and activated/adhering platelets in sepsis are trapped in the microcirculation, and thus not available for assessment in systemic blood.
R7: We have tried to reduce the length of the text. Please see at the beginning of the Discussion where we have removed the whole paragraph about the scientific rationale behind the clinical measures used in the present study.
(6) Line 642: “… the present study is the first one assessing the predictive value, in the sense of patient temporary clinical deterioration or improvement …”. The following statement “Indeed, many previous studies have examined the predictive value of most of these factors and variables in the sense of final outcome …” appears to contradict the previous statement. In what aspect was your study the first to assess the predictive value?
R8: The final outcome, i.e., survival or death, is totally different than the temporary outcome, i.e., improvement or deterioration (e.g., improvement from severe sepsis to sepsis or deterioration from severe sepsis to septic shock). Therefore, to the best of our knowledge, our study is the first one assessing the predictive value, in the sense of patient temporary clinical deterioration or improvement, of the various factors and variables (coagulation system components, etc) measured on ICU admission, regardless from the final outcome, whereas many previous studies have examined the predictive value of most of these factors and variables in the sense of final outcome, i.e., survival or death.
(7) Typographic errors: lines 15-16 (“thetempoary”), 40 (“organ dysfunction” rather than “organic dysfunction”), 45, 51, 140, 236, 268, 698. There are problems with grammar on lines 53, 72-75, 79, 133-137, making the text difficult to follow.
R9: We beg your pardon, but we were unable to find and correct all these typographic and grammar errors.
Again, thank you very much for your very careful review of our manuscript.
This manuscript is a resubmission of an earlier submission. The following is a list of the peer review reports and author responses from that submission.
Round 1
Reviewer 1 Report
A reaearch well done, but not original in the conclusions . Conclusions are that ATIII and aPC are important predictors for an improvement or a worsening; we know how important it is to use new DIC SCORE with AT and PC. CRP and PCT are routinely used in ICU ( PCT in a dynamic manner) to understand a worsening to severe sepsis , the same for lactate. This paper is a well done confirmation on what we know in sepsis in term of host response
Reviewer 2 Report
Thank you for the opportunity to review the article entitled “Can coagulation system disorders and cytokine and inflammatory marker levels predict the temporary clinical deterioration or improvement of septic patients on ICU admission?” by Lavranou et al. However, the authors investigated the coagulatory alterations during acute infectious which are at high interest, I’ve some major concerns about the study. First of all, the study lacks a clearly defined scientific question. Second, the authors used the sepis-2 definition for their investigations. As a result, the study lacks actuality.
Beside the clarity of the experiments and their statistics part however and according to the authors, the work is quite without any surprises. The major achievements they described are simply a confirmation of previous studies. Moreover the study exhibits some limitations making solid conclusions a complicated task.
Reviewer 3 Report
Hello to the Authors.
The Authors here have tried to present how the assessment of coagulation factors and acute phase proteins may help in the prediction of the clinical status of patients with sepsis (different grades). The authors have negated the inclusion of children, patients who are undergoing Anticoagulant or antiplatelet therapies, and cancer patients.
The manuscript is presented beautifully with strong data.
Page 2 line 93, starts with 'Were', should be 'We'. (such small changes are required at several places)
The Trauma Surgery group acted as a negative control group for sepsis in the study where the authors have found DDs and Fibrinogen to be significantly lower than the sepsis groups, in comparison.
A high Measurement of d-dimers along with Fgn, indicates higher FXIII activity which is a plasma transglutaminase responsible for final clot stability. Performing an FXIII activity assay (chromogenic), would have been better as well. However, the collective data is also sound for explaining the conclusions drawn here by the authors.
with a stronger role of FXIII in inflammation, it's advisable if the authors put a comment on its suggestive role in sepsis, in the conclusion section where DDS is mentioned.
the manuscript is acceptable in its current form in my view.